# Risk and adverse outcomes of fractures in patients with liver cirrhosis: two nationwide retrospective cohort studies

Ta-Liang Chen,[1,2,3] Chao-Shun Lin,[1,2,3] Chun-Chuan Shih,[4,5] Yu-Feng Huang,[6] Chun-Chieh Yeh,[7,8] Chih-Hsing Wu,[9] Yih-Giun Cherng,[3,10] Chien-Chang Liao[1,2,3,10,11]

## ABSTRACT

**Objective** The aim of this study is to evaluate fracture risk and post-fracture outcomes in patients with and without liver cirrhosis (LC).

**Design** Retrospective cohort study and nested fracture cohort study.

**Setting** This study was based on Taiwan's National Health Insurance Research Database that included information on: (1) 3941 patients aged 20 years and older newly diagnosed with LC between 2000 and 2003; (2) 688290 hospitalised fracture patients aged 20 years and older between 2006 and 2013.

**Primary and secondary outcome measures** Followed-up events of fracture from 2000 to 2008 were noted from medical claims to evaluate adjusted hazard ratios (HRs) and 95% confidence intervals (CIs) of fracture associated with LC. Adjusted odds ratios (ORs) and 95% CIs of adverse events after fracture were compared among patients with and without LC

**Results** The incidences of fracture for people with and without LC were 29.1 and 17.2 per 1000 person-years, respectively. Compared with controls, the adjusted HR of fracture was 1.83 (95% CI 1.67 to 2.01) for patients with LC. Previous LC was associated with risks of septicaemia (OR 1.77, 95% CI 1.60 to 1.96), acute renal failure (OR 1.63, 95% CI 1.33 to 1.99), and 30-day in-hospital mortality (OR 1.61, 95 %CI 1.37 to 1.89) after fracture.

**Conclusion** LC was associated with higher risk of fracture; patients with LC in particular had more complications and 30-day in-hospital mortality after fracture. Fracture prevention and attention to post-fracture adverse events are needed for these susceptible populations.

For numbered affiliations see end of article.

**Correspondence to**
Professor Chien-Chang Liao;
jacky48863027@yahoo.com.tw

## INTRODUCTION

Liver cirrhosis (LC) is the fourth most common cause of death in Europe, and causes more than one million deaths every year worldwide.[1 2] Prevalence of LC increased 68% during the 1992–2001 decade in the UK; in the US, it remains a pandemic chronic disease with more than 600000 patients in 1999–2010 whose economic burden was shown to have doubled during 1998 and 2003.[3–5] Although the epidemiology, pathogenesis, prevention and treatment of LC have

been studied,[6] complications of LC are not well understood.

## Strengths and limitations of this study

▶ Our study is a longitudinal retrospective cohort with large sample size.
▶ We used frequency matching and multivariate Cox proportional hazard models to control the confounding bias.
▶ This is the first study with comprehensive assessment of the impact of liver cirrhosis on fracture risk and outcomes.
▶ Our data lack clinical risk scores, lesion characteristics, biochemical measures, and lifestyles of patients with fracture and patients with liver cirrhosis.
▶ Detailed lifestyle data including smoking, alcohol drinking, and physical activity were not available.

Fracture causes disability and mortality, and places an economic burden on societies.[7 8] Since many studies found patients with LC had increased risk of osteoporosis,[9–11] fall and fracture were considered complications for patients with LC.[12–20] However, these studies did not compare the risk of fracture or falls in people with and without LC. Small sample size,[12 14 16–18] inadequate control for confounding factors,[12–18] case-control study design,[12 14 16–19] focus on specific population,[12] and lack of a control group[14 16 18 19] and of subgroup analysis[14–17 19 20] limited previous investigations. Information was also lacking on whether LC is associated with post-fracture adverse outcomes.

With the use of reimbursement claims from Taiwan's National Health Insurance programme, we conducted two nationwide cohort studies. The retrospective cohort study seeks to validate the risk of fracture in patients with LC. Whether LC was associated with adverse outcomes after fracture was reported in the nested fracture cohort study.

## METHODS
### Source of data
This study used the claims data of Taiwan's National Health Insurance Programme, which was implemented in March 1995 and covers 99% of 23 million people nationwide. The National Health Research Institutes (NHRI) established the National Health Insurance Research Database to record all beneficiaries' information about inpatient and outpatient medical services. This includes basic patient demographics, physician's primary and secondary diagnoses, treatment procedures, prescribed medications, and medical expenditures. All contracted medical clinics, hospitals and medical centres are required to submit computerised claim documents for medical expenses. The validity of this database has been favourably evaluated, and research articles based on it have been accepted in scientific journals worldwide.[21–23]

### Ethical approval
Insurance reimbursement claims used in this study were decoded with patients' identification scrambled for further research access. This study was conducted in accordance with the Helsinki Declaration. Although NHRI regulations do not require informed consent because patient identification has been decoded for privacy, this study was also approved by Taipei Medical University's Joint Institutional Review Board (TMU-JIRB-201705063; TMU-JIRB-201705084; TMU-JIRB-201506001; TMU-JIRB-201404070).

### Study design
This investigation included two studies. In study I (the retrospective cohort study), our purpose was to evaluate the risk of fracture for people with and without LC. From the representative sample of 1 000 000 insurance enrollees, we required at least two visits for medical care and a physician's primary diagnosis of cirrhosis of the liver to identify a cohort of 3941 newly diagnosed adults aged ≥20 years in 2000–2003. Those with only one medical visit and a physician's diagnosis of LC were not considered as cases of cirrhosis in this study. The frequency-matching procedure (by age and sex) was used to select the cohort with no previous medical records of LC. Both LC and non-LC cohorts had no history of fracture between the index date (date of LC diagnosis) and 1 January 1996 (the starting date of the Taiwan's National Health Insurance Programme). That is to say, there was no recorded previous fracture from onset of the database (1996) until the date of enrollment in the study (2000–2003). The outcome of this retrospective cohort study was an incidence of fracture that was identified during the follow-up period from the index date until the end of 2008 for LC and non-LC cohorts.

In study II (the nested fracture cohort study), our purpose was to evaluate the outcomes after fracture in patients with fracture with and without a history of LC. Study II included 688290 hospitalised patients with fracture in 2004–2013; we identified 7854 patients with a history of LC (defined as at least two visits for medical care and a physician's primary diagnosis of LC) within 24 months pre-fracture. Thirty-day in-hospital mortality, septicaemia, and acute renal failure after fracture were considered post-fracture outcomes and were compared in patients with fracture with and without LC in the nested fracture cohort study.

### Measurements and definitions
The variables in studies I and II were defined and described as follows. Patients' age was calculated by the date of fracture admission. Low-income status was determined by the National Health Insurance Bureau, which validates claims for those who qualify for waived healthcare copayment. Following previous suggestions,[21 22] fracture-associated medications were also analysed; these included anxiolytics, antiepileptics, antipsychotics, antidepressants and oral steroids. We used the International Classification of Diseases, Ninth Revision, Clinical Modification (ICD-9-CM) and administration codes of Taiwan's National Health Insurance to examine physicians' diagnoses and medical services provided. We considered coexisting medical conditions within 24 months before fracture admission. We identified patients' medical conditions and complications such as mental disorders (ICD-9-CM 290–319), hypertension (ICD-9-CM 410–405), chronic obstructive pulmonary disease (ICD-9-CM 491, 492 and 496), diabetes (ICD-9-CM 250), ischaemic heart disease (ICD-9-CM 410–414), stroke (ICD-9-CM 430–438), hyperlipidaemia (ICD-9-CM 272.0, 272.1 and 272.2), heart failure (ICD-9-CM 428), renal dialysis (administration code D8, D9), Parkinson's disease (ICD-9-CM 332), liver cirrhosis (ICD-9-CM 571.2, 571.5 and 571.6), fracture (ICD-9-CM 800–829), sepsis (ICD-9-CM 038), and acute renal failure (ICD-9-CM 548). Types of fracture and injury included skull fracture (ICD-9-CM 800–804), fracture of neck and trunk (ICD-9-CM 805–809), fracture of upper limb (ICD-9-CM 810–819), and fracture of lower limb (ICD-9-CM 820–829). Thirty-day in-hospital mortality was the main outcome in the nested fracture cohort study. In study I, we identified coexisting medical conditions and medications at baseline (in the 2 years before the enrollment date) and follow-up period. In study II, we identified coexisting medical conditions and medications in the 2 years before fracture admission.

### Statistical analysis
In study I, the categorical data for cohorts with and without LC were analysed by $\chi^2$ tests. The adjusted hazard ratios (HRs) and confidence intervals (CIs) of fracture risk associated with LC were calculated using multiple Cox proportional hazard models, controlling for age, sex, low income, mental disorders, hypertension, chronic obstructive pulmonary disease, diabetes, ischaemic heart disease, stroke, hyperlipidaemia, congestive heart failure, renal dialysis, Parkinson's disease, anxiolytics, antipsychotics, antiepileptics, antidepressants, and oral steroids, as were associations between LC and fracture risk in men, women, and every age group.

Table 1 Sociodemographics, coexisting medical conditions, and medication use in people with and without liver cirrhosis

| | Non-liver cirrhosis (n=15 764) | | Liver cirrhosis (n=3941) | | p Value |
|---|---|---|---|---|---|
| | n | (%) | n | (%) | |
| Sex | | | | | 1.0000 |
| Female | 4980 | (31.6) | 1245 | (31.6) | |
| Male | 10 784 | (68.4) | 2696 | (68.4) | |
| Age, years | | | | | 1.0000 |
| 20–29 | 620 | (3.9) | 155 | (3.9) | |
| 30–39 | 1924 | (12.2) | 481 | (12.2) | |
| 40–49 | 3628 | (23.0) | 907 | (23.0) | |
| 50–59 | 3624 | (23.0) | 906 | (23.0) | |
| 60–69 | 3320 | (21.1) | 830 | (21.1) | |
| ≥70 | 2648 | (16.8) | 662 | (16.8) | |
| Low income | 362 | (2.3) | 211 | (5.3) | <0.0001 |
| Coexisting medical conditions | | | | | |
| Mental disorders | 3974 | (25.2) | 1437 | (36.5) | <0.0001 |
| Hypertension | 5248 | (33.3) | 1406 | (35.7) | 0.0046 |
| COPD | 2892 | (18.4) | 992 | (25.2) | <0.0001 |
| Diabetes | 2242 | (14.2) | 944 | (24.0) | <0.0001 |
| Ischaemic heart disease | 2623 | (16.6) | 788 | (20.0) | <0.0001 |
| Stroke | 941 | (6.0) | 338 | (8.6) | <0.0001 |
| Hyperlipidaemia | 1468 | (9.3) | 294 | (7.5) | 0.0003 |
| Congestive heart failure | 453 | (2.9) | 255 | (6.5) | <0.0001 |
| Renal dialysis | 123 | (0.8) | 112 | (2.8) | <0.0001 |
| Parkinson's disease | 255 | (1.6) | 105 | (2.7) | <0.0001 |
| Medication use | | | | | |
| Anxiolytics | 5756 | (36.5) | 2550 | (64.7) | <0.0001 |
| Antipsychotics | 1559 | (9.9) | 805 | (20.4) | <0.0001 |
| Antiepileptics | 1547 | (9.8) | 773 | (19.6) | <0.0001 |
| Antidepressants | 1656 | (10.5) | 743 | (18.9) | <0.0001 |
| Oral steroids | 2549 | (16.2) | 672 | (17.1) | 0.1806 |

COPD, chronic obstructive pulmonary disease.

In study II, we used $\chi^2$ tests to examine other sociodemographic factors and medical conditions in hospitalised patients with fracture with and without a history of LC. Multiple logistic regressions were used to calculate adjusted odds ratios (ORs) and 95% CIs of 30-day in-hospital mortality, sepsis and acute renal failure after fracture associated with a history of LC, controlling for age, sex, low income, mental disorders, hypertension, diabetes, chronic obstructive pulmonary disease, ischaemic heart disease, stroke, congestive heart failure, Parkinson's disease, renal dialysis, hyperlipidaemia, and types of fracture.

## RESULTS

In study I, there was no significant difference in age and sex between cohorts with and without LC because we used a frequency-matching procedure (table 1). The LC cohort had more medical conditions than the non-LC cohort; these included mental disorders (p<0.0001), hypertension (p=0.0046), chronic obstructive pulmonary disease (p<0.0001), diabetes (p<0.0001), ischaemic heart disease (p<0.0001), stroke (p<0.0001), hyperlipidaemia (p=0.0003), heart failure (p<0.0001), renal dialysis (p<0.0001) and Parkinson's disease (p<0.0001). The use of anxiolytics (p<0.0001), antiepileptics (p<0.0001), antipsychotics (p<0.0001) and antidepressants (p<0.0001) was also higher in patients with LC than in people without LC.

During the 5–8 years of follow-up (study I), the incidence of fracture for cohorts with and without LC was 29.1 and 17.2 per 1000 person-years, respectively (table 2). The increased risk of fracture was found in the LC cohort

**Table 2**  Risk of fracture events for cohorts with and without liver cirrhosis[*]

| | People without liver cirrhosis | | | | People with liver cirrhosis | | | | Risk of fracture | |
|---|---|---|---|---|---|---|---|---|---|---|
| | n | Person-years | Events | Incidence† | n | Person-years | Events | Incidence† | HR | (95% CI)‡ |
| All | 15764 | 95430 | 1641 | 17.2 | 3941 | 23221 | 675 | 29.1 | 1.83 | (1.67 to 2.01) |
| Female | 4980 | 29405 | 718 | 24.4 | 1245 | 7352 | 249 | 33.9 | 1.53 | (1.32 to 1.78) |
| Male | 10784 | 66025 | 923 | 14.0 | 2696 | 15869 | 426 | 26.8 | 2.04 | (1.81 to 2.31) |
| Age, 20–39 years | 2544 | 15955 | 183 | 11.5 | 636 | 3838 | 97 | 25.3 | 2.12 | (1.59 to 2.81) |
| Age, 40–49 years | 3628 | 22574 | 295 | 13.1 | 907 | 5601 | 142 | 25.4 | 2.04 | (1.63 to 2.54) |
| Age, 50–59 years | 3624 | 21828 | 332 | 15.2 | 906 | 5442 | 124 | 22.8 | 1.67 | (1.34 to 2.07) |
| Age, 60–69 years | 3320 | 20111 | 373 | 18.5 | 830 | 4903 | 144 | 29.4 | 1.69 | (1.38 to 2.06) |
| Age, ≥70 years | 2648 | 14963 | 458 | 30.6 | 662 | 3437 | 168 | 48.9 | 1.70 | (1.41 to 2.03) |

*In the subgroup analysis, the HRs of skull fracture, neck or trunk fracture, fracture of upper limb, fracture of lower limb, and hip fracture associated with liver cirrhosis were 2.28 (95% CI 1.66 to 3.14), 1.75 (95% CI 1.44 to 2.12), 1.78 (95% CI 1.52 to 2.08), 1.94 (95% CI 1.65 to 2.27), and 2.22 (95% CI 1.70 to 2.89), respectively.
†Per 1000 person-years.
‡People with liver cirrhosis vs people without liver cirrhosis; Cox proportional hazard model controlling for all covariates listed in table 1.
CI, confidence interval; HR, hazard ratio.

after adjustment (HR 1.83, 95% CI 1.67 to 2.01). We also found significant associations between fracture risk and LC in men (HR 2.04, 95% CI 1.81 to 2.31), women (HR 1.53, 95% CI 1.32 to 1.78), and people aged 20–39 (HR 2.12, 95% CI 1.59 to 2.81), 40–49 (HR 2.04, 95% CI 1.63 to 2.54), 50–59 (HR 1.67, 95% CI 1.34 to 2.07), 60–69 (HR 1.69, 95% CI 1.38 to 2.06), and ≥70 years (HR 1.70, 95% CI 1.41 to 2.03).

Study II included 688560 hospitalised patients with fracture (table 3). Patients with fracture and LC included lower proportions of young adults (p<0.0001) but higher proportions of men (p<0.0001) and those with low-income status (p<0.0001). More patients with a history of mental disorders, diabetes, chronic obstructive pulmonary disease, ischaemic heart disease, stroke, congestive heart failure, Parkinson's disease, and renal dialysis were found in the LC cohort than in the control group (p<0.05 for all). In patients with fracture and previous LC, there were higher proportions of neck or trunk fracture (p<0.0001) and lower limb fracture (p<0.0001).

In study II, higher risks for post-fracture sepsis (OR 1.77, 95% CI 1.60 to 1.96), acute renal failure (OR 1.63, 95% CI 1.33 to 1.99), and 30-day in-hospital mortality (OR 1.61, 95% CI 1.37 to 1.89) were associated with previous LC (table 4). Patients wiht fracture and LC had a higher mean length of hospital stay (9.6±10.5 vs 8.5±13.8 days, p<0.0001) and medical expenditure (US$2500±2743 vs US$2212±2550, p<0.0001) than patients without LC.

In study II (table 5), previous LC was associated with higher risks of adverse events after fracture in women (OR 2.00, 95% CI 1.73 to 2.30), men (OR 1.61, 95% CI 1.45 to 1.80), and patients aged 30–39 (OR 1.84, 95% CI 1.35 to 2.51), 40–49 (OR 2.29, 95% CI 1.87 to 2.81), 50–59 (OR 1.90, 95% CI 1.53 to 2.36), 60–69 (OR 1.99, 95% CI 1.60 to 2.49), and ≥70 years (OR 1.41, 95% CI 1.24 to 1.61). Associations between post-fracture adverse events and LC

were significant in patients with fracture receiving surgery (OR 2.16, 95% CI 1.91 to 2.44), skull fracture (OR 1.60, 95% CI 1.35 to 1.90), or fracture of upper limb (OR 1.78, 95% CI 1.45 to 2.18), lower limb (OR 1.79, 95% CI 1.60 to 2.00), neck or trunk (OR 1.42, 95% CI 1.16 to 1.73), and hip fracture (OR 1.71, 95% CI 1.23 to 2.37).

Study II showed that alcohol dependence syndrome (OR 1.97, 95% CI 1.58 to 2.45), jaundice (OR 3.15, 95% CI 2.42 to 4.10), ascites (OR 2.49, 95% CI 2.08 to 2.98), gastrointestinal haemorrhage (OR 2.09, 95% CI 1.79 to 2.45), and hepatic coma (OR 2.77, 95% CI 2.27 to 3.38) were significant determinants for post-fracture adverse events (table 6). The risk of post-fracture adverse events increased when the number of cirrhotic indicators increased (OR 2.94, 95% CI 1.64 to 5.29).

## DISCUSSION

In study I (the retrospective cohort study), we observed a significant association between LC and fracture, with an 83% increased risk of fracture in patients with LC during the 5–8 years of follow-up. In study II (the nested fracture cohort study), we observed that patients with LC had significantly higher post-fracture complications and 30-day in-hospital mortality. Clinical indicators of the severity of LC, such as alcohol dependence syndrome, jaundice, ascites, gastrointestinal haemorrhage and hepatic coma, were all associated with more post-fracture adverse events.

In general, the prevalence of LC was higher in men than in women.[3 4] Older age and low socioeconomic status were factors associated with higher risk of LC.[3 23 24] Older age, male sex and low income were also risk factors for fracture, and the multivariate Cox proportional models was used to control these potential confounding effects when analysing the association between LC and

**Table 3** Characteristics of patients with fracture with and without liver cirrhosis

| | Liver cirrhosis | | | | |
| --- | --- | --- | --- | --- | --- |
| | No (n=680 706) | | Yes (n=7584) | | |
| | n | (%) | n | (%) | p Value |
| Age, years | | | | | <0.0001 |
| 20–29 | 78 164 | (11.5) | 82 | (1.1) | |
| 30–39 | 71 008 | (10.4) | 718 | (9.5) | |
| 40–49 | 83 951 | (12.3) | 1433 | (18.9) | |
| 50–59 | 110 315 | (16.2) | 1400 | (18.5) | |
| 60–69 | 96 376 | (14.2) | 1136 | (15.0) | |
| ≥70 | 240 892 | (35.4) | 2815 | (37.1) | |
| Sex | | | | | <0.0001 |
| Female | 330 440 | (48.5) | 2586 | (34.1) | |
| Male | 350 266 | (51.5) | 4998 | (65.9) | |
| Low income | 23 243 | (3.4) | 607 | (8.0) | <0.0001 |
| Medical centre | 181 324 | (26.6) | 1585 | (20.9) | <0.0001 |
| Coexisting medical conditions | | | | | |
| Mental disorders | 102 874 | (15.1) | 2027 | (26.7) | <0.0001 |
| Hypertension | 135 984 | (20.0) | 1604 | (21.2) | 0.0111 |
| Diabetes | 82 560 | (12.1) | 1543 | (20.4) | <0.0001 |
| COPD | 55 169 | (8.1) | 876 | (11.6) | <0.0001 |
| Ischaemic heart disease | 47 174 | (6.9) | 650 | (8.6) | <0.0001 |
| Stroke | 26 507 | (3.9) | 512 | (6.8) | <0.0001 |
| Congestive heart failure | 13 287 | (2.0) | 359 | (4.7) | <0.0001 |
| Parkinson's disease | 17 991 | (2.6) | 230 | (3.0) | 0.0355 |
| Renal dialysis | 7564 | (1.1) | 230 | (3.0) | <0.0001 |
| Hyperlipidaemia | 22 779 | (3.4) | 120 | (1.6) | <0.0001 |
| Type of fracture | | | | | |
| Fracture with surgery | 482 458 | (70.9) | 4040 | (53.3) | <0.0001 |
| Skull fracture | 119 165 | (17.5) | 1263 | (16.7) | 0.0520 |
| Fracture of neck and trunk | 97 677 | (14.4) | 1500 | (19.8) | <0.0001 |
| Fracture of upper limb | 256 356 | (37.7) | 2285 | (30.1) | <0.0001 |
| Fracture of lower limb | 322 161 | (47.3) | 3807 | (50.2) | <0.0001 |
| Hip fracture | 172 592 | (25.4) | 2551 | (33.6) | <0.0001 |
| Open fractures | 49 783 | (7.3) | 432 | (5.7) | <0.0001 |

COPD, chronic obstructive pulmonary disease.

fracture risk.[25] Furthermore, we found that the association between LC and fracture risk remained significant in every age group and in both sexes. The significant impact of LC on post-fracture adverse events was noted in men, women, in various age groups, and in people with various types of fracture. This phenomenon revealed the possible causal inference that LC was associated with fracture risk and post-fracture adverse events from the viewpoint of epidemiology. These findings are crucial because several previous studies were limited by focusing on specific populations and failed to investigate the association in a subgroup analysis.[14–17 19 20]

Mental disorders, hypertension, chronic obstructive pulmonary disease, diabetes, ischaemic heart disease, stroke, hyperlipidaemia, congestive heart failure, renal dialysis and Parkinson's disease were considered as coexisting medical conditions that were also fracture risk factors.[21 22 26–30] Confounding bias may have occurred in previous studies that lacked multivariate adjustment for these fracture-related and/or cirrhosis-related medical conditions.[12–18] Therefore, we used multiple Cox proportional hazard and multiple logistic regression models to control the confounding effects of medical conditions

**Table 4** Adverse events after fracture in patients with and without liver cirrhosis

|  | No LC, % | LC, % | OR (95% CI)* |
|---|---|---|---|
| 30-day in-hospital mortality | 1.2 | 2.2 | 1.61 (1.37 to 1.89) |
| Septicaemia | 2.5 | 5.5 | 1.77 (1.60 to 1.96) |
| Acute renal failure | 0.7 | 1.4 | 1.63 (1.33 to 1.99) |
| Medical expenditure, US$† | 2212±2550 | 2500±2743 | 158 (105 to 211) |
| Length of hospital stay, days† | 8.5±13.8 | 9.6±10.5 | 0.22 (0.08 to 0.52) |

*Controlled for all covariates listed in table 3.
†t-test showed mean±SD, p<0.0001 for medical expenditure and length of hospital stay; β coefficients and 95% CIs for medical expenditure and length of hospital stay associated with liver cirrhosis were calculated using multiple linear regression.
CI, confidence interval; LC, liver cirrhosis; OR, odds ratio.

when investigating the risks and outcomes of fracture in patients with LC in studies I and II.

Unlike previous investigations, we studied the impact of LC on post-fracture outcomes such as septicaemia, acute renal failure and mortality.[12–20] Patients with fracture and a history of LC had longer hospital stay and increased medical expenditure than people without LC in the nested fracture cohort study. Patients with LC had circulatory dysfunction and poor immune systems that compromised systemic inflammatory response and made them prone to renal failure and septicaemia,[23 31] particularly those patients with cirrhotic indicators such as alcohol dependence syndrome, jaundice, ascites, gastrointestinal haemorrhage, and hepatic coma. Therefore, higher mortality and increased use of medical resources might be encountered in the LC population during fracture admission.

There are several possible explanations for associations between LC and fracture risk. First, many studies found that patients with LC had increased risk of osteoporosis,[9–11] a condition that is an important determinant for fracture.[8] Fracture due to bone loss and the pathogenesis of osteoporosis among patients with LC is complex and multifactorial, and the exact mechanism remains uncertain. A previous study showed patients with cirrhosis and osteoporosis had lower levels of insulin-like growth factor 1 than patients with cirrhosis without osteoporosis.[32] Insulin-like growth factor 1 plays a major role in bone remodelling and maintenance of bone mass, and was found to be reduced in advanced cirrhosis.[33] In patients with cirrhosis, hyperbilirubinemia has also been shown to impair osteoblast proliferation, resulting in decreased bone formation and possibly accounting for the increased risk of fracture.[34] Second, corticosteroids

**Table 5** Liver cirrhosis associated with post-fracture adverse events in the stratification analysis by age, sex and type of fracture

|  | People without liver cirrhosis | | | People with liver cirrhosis | | | Risk of events* |
|---|---|---|---|---|---|---|---|
|  | n | Events* | Incidence | n | Events* | Incidence | OR (95% CI)† |
| Female | 330 440 | 11 082 | 3.4 | 2586 | 219 | 8.5 | 2.00 (1.73 to 2.30) |
| Male | 350 266 | 15 863 | 4.5 | 4998 | 402 | 8.0 | 1.61 (1.45 to 1.80) |
| Age, 20–29 years | 78 164 | 1862 | 2.4 | 82 | 6 | 7.3 | 2.34 (1.00 to 5.52) |
| Age, 30–39 years | 71 008 | 1698 | 2.4 | 718 | 47 | 6.6 | 1.84 (1.35 to 2.51) |
| Age, 40–49 years | 83 951 | 2215 | 2.6 | 1433 | 118 | 8.2 | 2.29 (1.87 to 2.81) |
| Age, 50–59 years | 110 315 | 3053 | 2.8 | 1400 | 99 | 7.1 | 1.90 (1.53 to 2.36) |
| Age, 60–69 years | 96 376 | 3362 | 3.5 | 1136 | 92 | 8.1 | 1.99 (1.60 to 2.49) |
| Age, ≥70 years | 240 892 | 14 755 | 6.1 | 2815 | 259 | 9.2 | 1.41 (1.24 to 1.61) |
| Fracture with surgery | 482 458 | 12 709 | 2.6 | 4040 | 297 | 7.4 | 2.16 (1.91 to 2.44) |
| Skull fracture | 119 165 | 8617 | 7.2 | 1263 | 157 | 12.4 | 1.60 (1.35 to 1.90) |
| Neck and trunk fracture | 97 677 | 4864 | 5.0 | 1500 | 107 | 7.1 | 1.42 (1.16 to 1.73) |
| Upper limb fracture | 256 356 | 4706 | 1.8 | 2285 | 104 | 4.6 | 1.78 (1.45 to 2.18) |
| Lower limb fracture | 322 161 | 14 467 | 4.5 | 3807 | 351 | 9.2 | 1.79 (1.60 to 2.00) |
| Hip fracture | 172 592 | 10 059 | 5.8 | 2551 | 265 | 10.4 | 1.71 (1.23 to 2.37) |

*Any adverse events included 30-day in-hospital mortality, septicaemia, and acute renal failure.
†Controlled for all covariates listed in table 3.
CI, confidence interval; OR, odds ratio.

**Table 6** Cirrhosis-related clinical indicators' effect on the outcomes of fracture admission in patients with liver cirrhosis

| Characteristics of cirrhosis before fracture admission | 30-day in-hospital adverse events* | | | |
|---|---|---|---|---|
| | n | Events | Incidence, % | OR (95% CI)† |
| No LC | 680 706 | 26 945 | 4.0 | 1.00 (reference) |
| Effects of liver admission | | | | |
| LC without liver admission | 5068 | 401 | 7.9 | 1.65 (1.49 to 1.84) |
| LC with liver admission | 2516 | 220 | 8.7 | 1.91 (1.65 to 2.20) |
| Effects of ADS | | | | |
| LC without ADS | 6397 | 530 | 8.3 | 1.70 (1.55 to 1.86) |
| LC with ADS | 1187 | 91 | 7.7 | 1.97 (1.58 to 2.45) |
| Effects of jaundice | | | | |
| LC without jaundice | 7013 | 554 | 7.9 | 1.64 (1.50 to 1.80) |
| LC with jaundice | 571 | 67 | 11.7 | 3.15 (2.42 to 4.10) |
| Effects of ascites | | | | |
| LC without ascites | 6337 | 479 | 7.6 | 1.59 (1.45 to 1.75) |
| LC with ascites | 1247 | 142 | 11.4 | 2.49 (2.08 to 2.98) |
| Effects of GI haemorrhage | | | | |
| LC without GI haemorrhage | 5688 | 439 | 7.7 | 1.62 (1.46 to 1.79) |
| LC with GI haemorrhage | 1896 | 182 | 9.6 | 2.09 (1.79 to 2.45) |
| Effects of hepatic coma | | | | |
| LC without hepatic coma | 6587 | 503 | 7.6 | 1.59 (1.45 to 1.75) |
| LC with hepatic coma | 997 | 118 | 11.8 | 2.77 (2.27 to 3.38) |
| Number of cirrhotic indicators‡ | | | | |
| 0 | 3772 | 254 | 6.7 | 1.36 (1.19 to 1.54) |
| 1 | 2378 | 206 | 8.7 | 1.83 (1.58 to 2.12) |
| 2 | 944 | 105 | 11.1 | 2.65 (2.15 to 3.27) |
| 3 | 359 | 43 | 12.0 | 2.95 (2.12 to 4.09) |
| ≥4 | 131 | 13 | 9.9 | 2.94 (1.64 to 5.29) |

*Adverse events included 30-day in-hospital mortality, septicaemia, and acute renal failure.
†Controlled for all covariates listed in table 3.
‡Liver-related illnesses included alcohol dependence syndrome, ascites, jaundice, gastrointestinal haemorrhage, and hepatic coma.
ADS, alcohol dependence syndrome; CI, confidence interval; GI, gastrointestinal; LC, liver cirrhosis; OR, odds ratio.

are frequently used in patients with autoimmune hepatitis and other inflammatory disorders. Even budesonide, a corticosteroid with minimal systemic availability, might lead to accelerated bone loss in patients with cirrhosis and postmenopausal women.[35] We postulated that medications used in the treatment of LC could also have an adverse effect on bone and calcium mobilisation and subsequent osteoporosis. Third, hepatic coma, poor cognitive function and psychiatric illness may play roles in the association between LC and risk of fracture.[25 36] Although hepatic encephalopathy does not commonly occur in patients with LC, its contribution to falls should not be ignored.[37]

Some study limitations need to be considered when interpreting the results. First, this study used retrospective reimbursement claims, which lack data on severity of LC, lifestyle factors, personal characteristics and biochemical data. Compared with the previous study,[19] theunavailable information is an important source of

bias. Second, since the patients were selected based on diagnoses from hospital inpatient care registers, patients with minor LC but no symptoms might not consult medical services, leading to underestimation of fracture risk in patients with LC because some cases with minor LC may have been in the non-LC group. Third, because our results are based on the data from Taiwan's National Health Insurance, the findings of this study could not be directly generalised to other populations.

## Conclusion

Our two cohort studies provide population-based evidence that LC is an important risk factor for fracture. We also note that patients with fracture and various clinical indicators of LC severity face increased risks of post-fracture adverse events. We presented risk factor analysis and a variety of clinical suggestions, including prevention, risk assessment and outcome-related information in patients with fracture and LC. Strategies

to prevent fracture and meticulous care to reduce post-fracture adverse events should be routinely considered for this population.

## Author affiliations

[1]Department of Anesthesiology, Taipei Medical University Hospital, Taipei, Taiwan
[2]Anesthesiology and Health Policy Research Center, Taipei Medical University Hospital, Taipei, Taiwan
[3]Department of Anesthesiology, School of Medicine, College of Medicine, Taipei Medical University, Taipei, Taiwan
[4]The School of Chinese Medicine for Post-Baccalaureate, I-Shou University, Kaohsiung, Taiwan
[5]Ph.D. Program for Clinical Drug Discovery from Botanical Herbs, Taipei Medical University, Taipei, Taiwan
[6]Department of Anesthesiology, Taitung Mackay Memorial Hospital, Taitung, Taiwan
[7]Department of Surgery, China Medical University Hospital, Taichung, Taiwan
[8]Department of Surgery, University of Illinois, Chicago, Illinois, USA
[9]Department of Family Medicine, National Cheng Kung University Hospital, Tainan, Taiwan
[10]Department of Anesthesiology, Shuan Ho Hospital, Taipei Medical University, Taipei, Taiwan
[11]School of Chinese Medicine, College of Chinese Medicine, China Medical University, Taichung, Taiwan

**Acknowledgements** This study is based in part on data obtained from the National Health Insurance Research Database provided by the Bureau of National Health Insurance, Ministry of Health and Welfare, and managed by the National Health Research Institutes. The interpretation and conclusions contained herein do not represent those of the Bureau of National Health Insurance, Ministry of Health and Welfare, or National Health Research Institutes.

**Contributors** All the authors revised and approved the contents of the submitted article. TLC and CCL created the idea of the manuscript and wrote the draft. CCL conducted statistical analysis of data. All the authors made substantial contributions to interpretation of data and carried out a critical revision of the manuscript for important intellectual content.

**Funding** This research was supported in part by Shuang Ho Hospital, Taipei Medical University (104TMU-SHH-23), Taiwan's Ministry of Science and Technology (MOST105-2629-B-038-001; MOST105-2314-B-038-025; MOST104-2314-B-038-027-MY2; NSC102-2314-B-038-021-MY3), and Taiwan's Ministry of Health and Welfare Clinical Trial and Research Center of Excellence (MOHW105-TDU-B-212-133019).

**Competing interests** None declared.

**Patient consent** This study was conducted inaccordance with the Helsinki Declaration. Informed consent is not required because patient identification has been decoded forprivacy.

**Ethics approval** Taipei Medical University's Joint Institutional Review Board (TMU-JIRB-201404070).

**Provenance and peer review** Not commissioned; externally peer reviewed.

**Data sharing statement** Data sharing statement: dataset available from the Taiwan's National Health Research Institutes (http://nhird.nhri.org.tw/index1.php).

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
