## [Reviewer comments · BMJ Open]

ARTICLE DETAILS

TITLE (PROVISIONAL)	Risk and adverse outcomes of fractures in patients with liver cirrhosis: two nationwide retrospective cohort studies
AUTHORS	Chen, Ta-Liang; Lin, Chao-Shun; Shih, Chun-Chuan; Huang, Yu-Feng; Yeh, Chun-Chieh; Wu, Chih-Hsing; Cherng, Yih-Giun; Liao, Chien-Chang

VERSION 1 – REVIEW

REVIEWER	Stephen Ryder Nottingham University Hospitals NHS Trust UK
REVIEW RETURNED	05-May-2017

GENERAL COMMENTS	The authors describe the impact of a diagnosis of liver cirrhosis on fracture risk from health insurance records in Taiwan. I would make the following comments: 1. The methods section is inadequate to understand what was done and why. The authors should separate out the two studies and describe the methods used in each clearly.2. What is meant by "representative sample of 1,000,000 enrollees" (p11 line 7)? Assuming this is the cirrhosis diagnosis why take a sub-population of the whole to study-you appear to have 23 million so what is the rationale for restricting to 1?3. Does line 18 mean "no recorded previous fracture from onset of database (1996) until the date of enrollment to the study (2000-2003)?" If so the way it is put is very difficult to follow and confuses things.4. Fractures at different sites may be better coded than others and there is the potential for ascertainment bias. It is well described that axial fractures are poorly coded in most clinical and insurance datasets. Patients with alcohol related liver disease have more admissions to hospital are more likely to have x rays of their spine carried out and acquire more fracture diagnoses. For these reasons I think it is imperative to have a much more detailed analysis of fracture by site and to at least state some measure of hospitalisation rates in the cohorts. Hip fracture is usually very well coded as all require surgery so this may be the best real measure but the more detail on fracture site that is available the better. I note the authors sub divide by upper/lower limb in some of the tables and text but I do
---

	not feel this is adequate and should include clear definitions of fractures where possible. If the dataset does not allow this the reasons should be clearly stated. 5. I don't understand what is meant by "a biological gradient with number of cirrhotic features" p14 line 7. 6. The cirrhosis group in the follow up study have multiple co-morbidities which increase fracture risk. The manuscript just states after correction the fracture rates remain significantly elevated but the degree of difference in the groups is very large. I think more explanation as to how this correction for comorbidities is done is required as the groups look dramatically different. 7. Do patients in the nested cohort study have multiple admissions with liver related codes prior to their fracture? It would help to know if fracture preceded the development of liver failure. I accept the data may not be able to fully answer the questions but it would be helpful in contextualising the risk if that is possible.
--	--

REVIEWER	Grace Su University of Michigan, USA
REVIEW RETURNED	25-May-2017

GENERAL COMMENTS	This interesting study uses the Taiwan National Health Insurance Research Database to identify two retrospective cohorts of 1) patients with newly diagnosed cirrhosis between 2000 and 2003 and examined the development of falls in the cohort in the followup period and 2) patients who had a hospitalized fracture between 2006 to 2013 and examined the 30 day hospital mortality. 1) Diagnosing liver cirrhosis in the 2 cohorts is a main variable- how was this diagnosis made in the insurance dataset for both cohorts and more importantly, what is the sensitivity and specificity of the methodology used. 2) How was medication use determined? If it is in the dataset then, the actual medications in each of the groups need to be provided in the appendix. 3) How was missing data handled? 4) How was death confirmed- is this captured in dataset? 5) Please specify codes or methods for differentiating different types of fractures. 6) Please be consistent with terminology of "nested fracture cohort" or "fracture nested cohort" 7) Page 9 stated the starting date of the Taiwan National Health Program was Jan 1, 1996 while page 8 stated the program was implemented March 1995
--

REVIEWER	Michalis Katsoulis Farr Institute/UCL
REVIEW RETURNED	12-Jun-2017

GENERAL COMMENTS	This is an interesting paper that focuses on the association between  a) liver cirrhosis and fractures b) liver cirrhosis and post fracture outcomes There are still some issues in relation to the study design and the statistical analysis that need to be clarified For the relationship between liver cirrhosis and fractures  i) please mention in the statistical analysis part that you adjust for all covariates in Table 1 ii) Were the coexisting medical conditions and the medication use measured at baseline, or during the follow-up time? Please clarify For the relationship between liver cirrhosis and post fracture outcomes; your sample consists of participants with fractures and you are investigating the risk of developing adverse events due to liver cirrhosis.  iii) please change the objective in the abstract and the title in table 4. You are comparing people with and without liver cirrhosis; your sample does not consist of participants with liver cirrhosis only. iv) In the logistic regression model, did you use age and coexisting medical conditions at the time of fracture event? Please clarify v) please use linear regression in table 4 for the relationship of Medical expenditure and Length of hospital stay with liver cirrhosis, so that we can incorporate other covariates as well. Calculate beta coef (95% CI) vi) you state that one of your outcomes is mortality, but in fact it is short-term (in-hospital) mortality (within 30 days). Please mention this in the abstract and the main text as well. vii) Finally, both in the introduction and the discussion, you emphasize on the problems of previous studies due to confounding, while you devote only one line for the confounding problems of your study. The lack of data on severity of LC, lifestyle factors, personal characteristics, and biochemical data are important source of bias. Please highlight this issue and contrast your limitation with the relative advantages of other studies.
---

VERSION 1 – AUTHOR RESPONSE

To reviewer #1:

1. Q: The methods section is inadequate to understand what was done and why. The authors should separate out the two studies and describe the methods used in each clearly.

Reply: By your comments, the descriptions of Methods and Results for the Study I and Study II were revised.

Thanks for the comments.

2. Q: What is meant by "representative sample of 1,000,000 enrollees" (p11 line 7)? Assuming this is the cirrhosis diagnosis why take a sub-population of the whole to study-you appear to have 23 million so what is the rationale for restricting to 1?

Reply: In fact, the program of Taiwan's National Health Insurance covered 23 million people in Taiwan. However, for protecting personal privacy, we could not use the whole reimbursement claims of Taiwan's National Health Insurance. Therefore, the Taiwan's National Health Research Institutes released "the representative sample consisted of one million persons" for researchers to do academic research. The one-million sample from Taiwan's National Health Insurance Research Database was used in many studies that were accepted in many important worldwide journals [Diabetes Care. 2014;37:2246-52] [Allergy. 2016;71(11):1626-1631] [Mayo Clin Proc. 2014;89(2):163-72] [Mayo Clin Proc. 2013;88(10):1091-8] [J Neurol Neurosurg Psychiatry. 2013;84(4):441-5] [J Neurol Neurosurg Psychiatry. 2012;83(12):1186-92].

Thank you for the understanding.

3. Q: Does line 18 mean "no recorded previous fracture from onset of database (1996) until the date of enrollment to the study (2000-2003)?" If so the way it is put is very difficult to follow and confuses things.

Reply: By your comments, we revised the description as "Both LC and non-LC cohorts had no history of fracture between the index date (date of LC diagnosis) and January 1, 1996 (the starting date of the Taiwan's National Health Insurance Program). That is to say, there was no recorded previous fracture from onset of database (1996) until the date of enrollment to the study (2000-2003)" in the Methods section.

Thanks for the comments.

4. Q: Fractures at different sites may be better coded than others and there is the potential for ascertainment bias. It is well described that axial fractures are poorly coded in most clinical and insurance datasets. Patients with alcohol related liver disease have more admissions to hospital are more likely to have x rays of their spine carried out and acquire more fracture diagnoses. For these reasons I think it is imperative to have a much more detailed analysis of fracture by site and to at least state some measure of hospitalisation rates in the cohorts. Hip fracture is usually very well coded as all require surgery so this may be the best real measure but the more detail on fracture site that is available the better. I note the authors sub divide by upper/lower limb in some of the tables and text but I do not feel this is adequate and should include clear definitions of fractures where possible. If the dataset does not allow this the reasons should be clearly stated.

Reply: The risk of hip fracture associated with liver cirrhosis was addressed in the footnotes of Table 2 (Study I). By your comments, we added the type of fracture "hip fracture" in the Study II. Because of adding hip fracture in the Table 3, the OR in Table 4, Table 5, and Table 6 were also revised.

5. Q: I don't understand what is meant by "a biological gradient with number of cirrhotic features" p14 line 7.

Reply: By your comments, the sentence was revised as "The risk of post-fracture adverse events increased with the number of cirrhotic indicators increased".

Thanks for the comments.

6. Q: The cirrhosis group in the follow up study have multiple co-morbidities which increase fracture risk. The manuscript just states after correction the fracture rates remain significantly elevated but the degree of difference in the groups is very large. I think more explanation as to how this correction for comorbidities is done is required as the groups look dramatically different.

Reply: In the Study I and Study II, we noticed that LC group had higher proportions of medical conditions than non-LC group. The confounding effects from medical conditions should be controlled in this manuscript. Thus, we used multivariate Cox proportional hazard regressions to control these confounding effects on the association between LC and fracture risk in Study I. We also used multivariate logistic regressions to control these confounding effects on the association between LC and pos-fracture adverse events in Study II. The above descriptions were stated clearly in the Methods and Discussion section.

By your comments, we revised the term "adjust" as "control". In the retrospective cohort study, the good procedures to control confounding effects are "matching" and "multivariate regression control". In our manuscript, frequency matching was used to balance the difference in age and sex. The confounding effects from medical conditions were controlled in the multivariate regressions models. The statements "Confounding bias may occurred in previous studies that lacked multivariate adjustment for these fracture-related and/or cirrhosis-related medical conditions.[12-18] Therefore, we used multiple Cox proportional hazard and multiple logistic regression models to control the confounding effects of medical conditions when investigating the risks and outcomes of fracture in patients with LC in Study I and Study II." were revised in the Discussion section.

Thanks for the comments.

7. Q: Do patients in the nested cohort study have multiple admissions with liver related codes prior to their fracture? It would help to know if fracture preceded the development of liver failure. I accept the data may not be able to fully answer the questions but it would be helpful in contextualising the risk if that is possible.

Reply: By your comments, we added the analysis "the effects of liver admission on the post-fracture outcomes". Please see Table 6. The corresponding descriptions were also added in the Results section.

Thanks for the comments.

To reviewer #2:

1. Q: Diagnosing liver cirrhosis in the 2 cohorts is a main variable- how was this diagnosis made in the insurance dataset for both cohorts and more importantly, what is the sensitivity and specificity of the methodology used.

Reply: To increase the likelihood of capturing patients with liver cirrhosis, the selection criteria were stated in the Methods as "at least two visits for medical care with the physician's primary diagnosis of cirrhosis of the liver". By your comments, we revised some descriptions in nested cohort study (Study II) in the Methods section for clarifying the selection criteria of cirrhotic cases and it was showed as "we identified 7854 with history of LC (defined as at least two visits for medical care with the

physician's primary diagnosis of liver cirrhosis) within pre-fracture 24 months". The criteria of "at least two visits for medical care with the physician's primary diagnosis of liver cirrhosis" were used in our previous studies that were accepted in important journals [Br J Surg. 2013;100(13):1784-90] [Atherosclerosis. 2017;263:29-35]. Although the criteria used in this manuscript may be a little strict (a little low sensitivity), we have much more confidence that all cirrhotic cases identified were reached to be 100% true (high specificity). In addition, non-cirrhosis group in our studies were clearly without any physician's diagnosis of cirrhosis. However, some people have little cirrhosis without clinical features may be included in non-cirrhosis group and this is also one of our study limitations. We also added some statements in the Discussion section.

Thanks for the comments.

2. Q: How was medication use determined? If it is in the dataset then, the actual medications in each of the groups need to be provided in the appendix.

Reply: The use of medications (included anxiolytics, antipsychotics, antiepileptics, antidepressants, and oral steroids) was identified before the enrollment date within 2 years and the follow-up period. By your comments, we added the above descriptions in the Methods section. The detailed medications were also listed in the appendix.

Thanks for the comments.

3. Q: How was missing data handled?

Reply: Because we used the reimbursement claims of Taiwan's National Health Insurance that was a strict system with automatic review process of reimbursement claims, there was no missing data in this study.

Thanks for the comments.

4. Q: How was death confirmed- is this captured in dataset?

Reply: The death was identified from the same database "Taiwan's National Health Insurance Research Database".

Thanks for the comments.

5. Q: Please specify codes or methods for differentiating different types of fractures.

Reply: By your comments, the descriptions of codes for different types of fractures were added in the Methods section.

Thanks for the comments.

6. Q: Please be consistent with terminology of "nested fracture cohort" or "fracture nested cohort"

Reply: By your comments, we revised the terminology as "nested fracture cohort study" throughout the manuscript.

Thanks for the suggestions.

7. Q: Page 9 stated the starting date of the Taiwan National Health Program was Jan 1, 1996 while page 8 stated the program was implemented March 1995

Reply: Yes, the Taiwan National Health Program was implemented since March 1, 1995, however, the reimbursement claims of Taiwan's National Health Insurance could be available since Jan 1, 1996. There was no incorrect description in the Methods.

Thanks for the concerns.

To reviewer #3:

1. Q: please mention in the statistical analysis part that you adjust for all covariates in Table 1

Reply: By your comments, the adjusted factors were mentioned in the Statistical analysis, please see the Methods section.

Thanks for the suggestions.

2. Q: Were the coexisting medical conditions and the medication use measured at baseline, or during the follow-up time? Please clarify For the relationship between liver cirrhosis and post fracture outcomes; your sample consists of participants with fractures and you are investigating the risk of developing adverse events due to liver cirrhosis.

Reply: By your comments, we added the descriptions in the Methods section as "In the Study I, we identified co-existing medical conditions and medications in the baseline (before the enrollment date within 2 years) and follow-up period. In the Study II, we identified co-existing medical conditions and medications before fracture admission within 2 years".

Thanks for the comments.

3. Q: please change the objective in the abstract and the title in table 4. You are comparing people with and without liver cirrhosis; your sample does not consist of participants with liver cirrhosis only.

Reply: By your comments, the objectives in the Abstract "The aim of this study is to evaluate fracture risk and post-fracture outcomes in patients with and without liver cirrhosis" and the title in Table 4 "Adverse events after fracture in patients with and without liver cirrhosis" were revised.

Thanks for the comments.

4. Q: In the logistic regression model, did you use age and coexisting medical conditions at the time of fracture event? Please clarify.

Reply: The logistic regression was used for Study II in the tables 4. The age was calculated by the date of fracture admission. The coexisting medical conditions were calculated within 24 months before fracture admission. By your comments, we added the above descriptions in the Methods section.

Thanks for the comments.

5. Q: please use linear regression in table 4 for the relationship of Medical expenditure and Length of hospital stay with liver cirrhosis, so that we can incorporate other covariates as well. Calculate beta coef (95% CI)

Reply: By your comments, the multivariate linear regression was used to calculate beta coefficient and 95% CI for liver cirrhosis associated with the length of stay and medical expenditure of fracture admission. Please see the Table 4 and the corresponding descriptions were also added in the Results section.

6. Q: you state that one of your outcomes is mortality, but in fact it is short-term (in-hospital) mortality (within 30 days). Please mention this in the abstract and the main text as well.

Reply: By your comments, all term “mortality” used in abstract and text were revised as “30-day in-hospital mortality”.

Thanks for the comments.

7. Q: Finally, both in the introduction and the discussion, you emphasize on the problems of previous studies due to confounding, while you devote only one line for the confounding problems of your study. The lack of data on severity of LC, lifestyle factors, personal characteristics, and biochemical data are important source of bias. Please highlight this issue and contrast your limitation with the relative advantages of other studies.

Reply: By your comments, we added the description “Compared with the previous well-adjustment study [Clin Gastroenterol Hepatol. 2015;13:1670-5], the above unavailable information is an important source of bias” In the study limitation.

Thanks for the comments.

VERSION 2 – REVIEW

REVIEWER	Grace Su University of Michigan USA
REVIEW RETURNED	05-Aug-2017

GENERAL COMMENTS	None
------

REVIEWER	Katsoulis Michail UCL, UK
REVIEW RETURNED	12-Aug-2017

GENERAL COMMENTS	The authors have successfully addressed my comments
---